# Adolescent Menstrual Health Literacy in Low, Middle and High-Income Countries: A Narrative Review

**DOI:** 10.3390/ijerph18052260

**Published:** 2021-02-25

**Authors:** Kathryn Holmes, Christina Curry, Tania Ferfolja, Kelly Parry, Caroline Smith, Mikayla Hyman, Mike Armour

**Affiliations:** 1Centre for Educational Research, Western Sydney University, Locked Bag 1797, Penrith, NSW 2751, Australia; C.Curry@westernsydney.edu.au (C.C.); 19168353@student.westernsydney.edu.au (S.); t.ferfolja@westernsydney.edu.au (T.F.); 2NICM Health Research Institute, Western Sydney University, Locked Bag 1797, Penrith, NSW 2751, Australia; Kelly.Parry@westernsydney.edu.au (K.P.); caroline.smith@westernsydney.edu.au (C.S.); mhyman124@gmail.com (M.H.); m.armour@westernsydney.edu.au (M.A.); 3Translational Health Research Institute (THRI), Western Sydney University, Locked Bag 1797, Penrith, NSW 2751, Australia

**Keywords:** menstrual health literacy, menstrual health education, menstrual hygiene management, menstruation, dysmenorrhea

## Abstract

Background: Poor menstrual health literacy impacts adolescents’ quality of life and health outcomes across the world. The aim of this systematic review was to identify concerns about menstrual health literacy in low/middle-income countries (LMICs) and high-income countries (HICs). Methods: Relevant social science and medical databases were searched for peer-reviewed papers published from January 2008 to January 2020, leading to the identification of 61 relevant studies. Results: A thematic analysis of the data revealed that LMICs report detrimental impacts on adolescents in relation to menstrual hygiene and cultural issues, while in HICs, issues related to pain management and long-term health outcomes were reported more frequently. Conclusions: In order to improve overall menstrual health literacy in LMICs and HICs, appropriate policies need to be developed, drawing on input from multiple stakeholders to ensure evidence-based and cost-effective practical interventions.

## 1. Introduction

Puberty is a challenging time for adolescents due to multiple social, physical and psychological changes [1]. The onset of menstruation is a particularly salient topic for female adolescents because it can have a significant impact on their education and overall quality of life during puberty and after [2]. This paper and the research on which it reports generally aligns menstruation with girls/women, reflecting the bulk of current literature on the topic. The researchers are aware that menstruation and menstrual issues are relevant to many individuals who do not necessarily identify within the narrow binary gender constructions girl/boy, woman/man and would like to acknowledge that not all people who menstruate identify as a girl/woman/female. The ability to manage menstrual health with dignity and have access to healthcare and adequate sanitation facilities without any stigma is every human’s right [3]. Despite the fact that almost 800 million people menstruate worldwide [4], in some cultures menstruation is still regarded as a taboo topic, shrouded in a culture of silence, with a lack of information, products and infrastructure to manage it [3,5,6]. Even in high-income countries (HICs) such as Australia, young women and their parents often report feeling uncomfortable discussing menstruation [7,8], leading to poor menstrual health literacy [9]. Adolescent girls are a vulnerable cohort since inadequate measures to support their menstrual health can have a profound impact on their confidence [10], education [2] and participation in daily activities [11].

The degree to which individuals can understand, obtain and process health information and services is defined as health literacy [12]. Menstrual health literacy is particularly important for those of school age, as it is during these years that adolescents will first experience menstruation. Girls in low and middle-income countries (LMICs) are often misinformed or uninformed about menstrual health literacy and thus underprepared for menarche [13]. Knowledge gaps can lead to shame and isolation as well as unhygienic practices during menstruation [13]. Despite the existence of a vast amount of literature dedicated to menstrual hygiene management in LMICs [10], only a small proportion of research focuses on menstrual health disorders and their management in adolescents. Adolescents have been found to suffer from high rates of period pain (dysmenorrhea) or painful uterine cramping during menstruation, which is a menstrual health issue [14,15,16]. Both menstrual hygiene and menstrual health issues can contribute to absenteeism from school and social activities amongst adolescents, leading them to refrain from social interaction and to rely on inadequate self-medication [17].

In the case of high-income countries (HICs), the research literature dedicated to menstrual health in adolescents is relatively limited, despite dysmenorrhea being common and usually poorly managed [18] and young women reporting a lack of support from schools and universities in relation to menstruation [14]. Dysmenorrhea results in a significant negative impact on quality of life, reduces the ability to perform normal daily activities [19], and commonly occurs with other bothersome symptoms such as emotional changes and fatigue [20]. For young women at school or university, period pain often has a significant impact on their academic performance [21]. The relative paucity of research in this area in HICs may reflect an assumption that ‘period poverty’ is not a significant issue in these countries; however, this is likely to be incorrect [22].

This review synthesizes the current literature dedicated to adolescent menstrual health literacy across low, middle and high-income countries. Given that the impact of poor menstrual health can have long-term ramifications due to missed schooling and other opportunities, it is vital to understand how this issue affects all young menstruators, regardless of geographical or economic status.

## 2. Materials and Methods

In order to identify relevant peer-reviewed journals and reports, a structured search was performed on research related to the menstrual health literacy of adolescents around the world. The search included literature published from January 2008 through to January 2020. The primary focus of the search was on the educational delivery of information about menstruation in schools. Specific topics included: identification of ‘normal’ menstruation, menstrual hygiene management, identification and treatment of menstrual health disorders (including primary and secondary dysmenorrhea, pre-menstrual symptoms, endometriosis and polycystic ovarian syndrome), attitudes towards menstrual health literacy and intervention studies in schools.

### 2.1. Search Strategy and Results

To identify relevant peer-reviewed journal articles, a database full text search was performed. Databases searched included: Directory of Open Access Journals (DOAJ), Elsevier, Gale Academic, Pro Quest, Sage, Taylor and Francis Online, Springer, EmeraldInsight, BioMedCentral, PubMed and Scopus. A general Google.com search was conducted to identify relevant policy documents and government and international body reports (World Bank, World Health Organization, United Nations Sustainable Development Goals and International Women’s Health Coalition).

Key terms used included: Menstrual health literacy in adolescents, Menstrual health literacy, Menstrual health education, Dysmenorrhea education, Menstrual hygiene management in schools, Menstrual health in schools, Menstruation in schools and Menstrual education in schools. Initial literature searches were combined, duplicates removed, and 150 articles were screened based on the inclusion criteria with 86 subsequently rejected from further review based on the exclusion criteria. A total of 64 studies were included in the final review. The search process flow and results are summarized in Figure 1.

For the purposes of this review, the findings have been grouped by country income level based on the World Bank Atlas method [23]. In line with our previous work in this area, LICs and MICs are grouped together into the categorization LMICs [2]. Fifty-two of the studies were conducted in LMICs, and twelve of the studies were conducted in HICs. 

### 2.2. Inclusion and Exclusion Criteria

Articles that were included related to the following topics:Delivery of menstrual health education in schools;Policies and reports related to menstrual health literacy in schools;Attitudes and biases towards menstrual health literacy impacting school-going adolescents;Assessing menstrual hygiene management within schools;Intervention studies of menstrual health education programs in schools;Dysmenorrhea education and treatment in schools.Articles that were excluded:Those that focused on menstrual health literacy beyond schools, i.e., colleges, universities and the general community;Those that were published before 2008.

### 2.3. Thematic Analysis 

Braun and Clarke (2006) devised a six-phase guide as a framework to be used for conducting qualitative analysis of data [24]. The method is not tied to a particular theoretical perspective, making it flexible in nature, suitable for the diversity of the studies that have been included for our review [25]. The first step involved familiarizing ourselves with the data collected via the search process. At this stage, initial impressions of the studies were noted, followed by the creation of a table that summarized the study design, location, age range of the participants, sample size, population and the primary findings. At the second stage, the initial coding of the data was conducted using open coding, and the codes were developed and modified as we worked through the different selected studies. At stage 3, the codes were examined to identify common themes across the data, and we formulated a set of initial themes (Table 1). At stage 4, the preliminary themes were further reviewed and narrowed down to reflect the broader nature of the purpose of our research, i.e., to gauge the approach to and the loopholes in menstrual health literacy, in schools, across low, middle and high-income countries. Stage 5 involved defining the final themes, and the data were categorized accordingly (Table 2) with further sub-themes/divisions in the data based on the location of the studies, i.e., a low, middle or high-income country. The last step involved writing up the review using the final three underlying themes revealed from the semantic thematic analysis:  Menstrual hygiene management; Menstrual health issues; Attitudes towards menstruation.

## 3. Results and Discussion

A total of 61 studies were included in this literature review (Figure 1). An aggregative synthesis was performed on the literature, and the data were organized in the form of a table (Appendix A) that contains the details of the studies including the country where they were conducted, the number and age range of participants, as well as the study design and population type.

Most studies (n = 40) focused on menstrual hygiene management, while 33 studies focused upon menstrual health issues and 39 studies focused upon attitudes towards menstruation. About two thirds of all studies focused on more than one topic, while only 22 studies focused upon a single issue. Eleven of those 22 studies that focused on a single issue related to menstrual health issues. Menstrual hygiene management and attitudes towards menstruation were the most common pairing, with 17 articles exclusively tackling those two themes. The included studies were conducted in schools with a participant age range of 10–26 years and where a focus on menstrual health literacy of adolescents was evident.

### 3.1. Menstrual Hygiene Management

#### Low and Middle-Income Countries (LMICs)

Menstrual hygiene was the most common issue focused upon by studies, with 40 studies examining menstrual hygiene. These studies were all based in LMICs. Poor menstrual hygiene in LMICs has been attributed, in part, to a lack of resources and materials required for appropriate hygiene, including the lack of provision of adequate sanitary materials such as soap for cleaning hands [26,27,28] and sanitary napkins [29,30]. Additionally, adolescents often lack appropriate WASH (Water and Sanitation for Health) infrastructure such as safe, gender-separated, accessible and clean toilets, designed to ensure privacy, as well as access to clean water. This issue was observed in studies within Nepal [31], Uganda [28], Cambodia [32], Zambia [26,33], India [30,34,35,36], Bangladesh [37], Kenya [38] and Philippines [39]. 

Another pitfall for adolescents in most LMICs was that the primary source of menstrual information for these girls was usually their mothers, who themselves had inadequate and insufficient transferrable knowledge due to low literacy levels [34,40,41,42,43,44,45,46,47] or were hesitant to talk to their daughters regarding the significance of hygienic practices and a healthy attitude towards menstruation [48]. There was evidence that the lack of prior knowledge about menstruation and menstrual management amongst adolescents when their first period began (menarche) could lead them to experience anxiety, shock and shame as they failed to understand what is happening to their bodies [48,49,50].

Adolescents in LMICs sometimes resorted to using pieces of old cloth and blankets torn into rags, or tissue/toilet paper or pieces of (bedding) mattress instead of disposable sanitary napkins, which can be attributed to a lack of menstrual hygiene education and awareness [26,27,35,36,40,46,49] or to low socioeconomic status/family income [27,28,32,33,50], or lack of adequate disposal facilities at school [31]. This phenomenon led to some students disposing of menstrual materials in the pit latrine [26], or burning/burying/throwing them in secluded places [50] or leaving them lying on the latrine floor, taking them home in a plastic bag [29], flushing them in the toilet [48] or keeping them in their pockets instead of putting them in the public trash facilities over fear of being ‘exposed’ [51].

In LMICs, poor menstrual hygiene amongst adolescent females has been found to interfere with students’ overall quality of school performance [52]. A study conducted in India revealed that 50% of 3617 adolescent females complained of an inability to concentrate when in school during menstruation, on account of embarrassment and anxiety due to fear of leakage and menstrual stains [36]. A similar observation was made in Uganda with more than half of the 205 participants of the study avoiding standing up to answer questions in class and having difficulty concentrating due to discomfort, distress and concerns about odour [53]. Additionally, lack of appropriate menstrual hygiene management also impacted absenteeism from school, with adolescents preferring to stay at home during their menstrual cycle [54].

Menstrual hygiene practices amongst adolescents can be influenced by the content and delivery of menstrual education programs in LMIC schools as evidenced by the success of interventional studies conducted in Pakistan [29], Ethiopia [55], Bangladesh [49] and Nepal [45]. UNICEF Pakistan worked with six government high schools in an attempt to improve menstrual hygiene management outcomes with adolescents. They attempted to improve WASH facilities, distribute sanitary material and provide information booklets regarding menstruation to teachers and adolescent girls, collectively termed as the Learning-Acting-Learning (LAL) approach, which ultimately showed significant improvement in menstrual hygiene outcomes in schools after six weeks of implementation [29]. Despite this fact, relatively few menstrual health education programs are implemented [37,40], and the ones that are lack information regarding the functions of reproductive organs [40,50], appropriate hygiene management [36,37,49,56] and menstrual health disorders and their treatment [56,57,58,59]. At the same time, as revealed by Blake et al. (2018), distribution of a menstrual health booklet named ‘Growth & Changes’ was not enough to significantly alter the menstrual health outcomes for adolescents, since a significant change requires radical governmental policy change and involvement, as well as an attitude shift in the community [55].

Issues related to menstrual hygiene management appear to be prevalent in LMICs, and by comparison, they were not reported in studies conducted in HICs. This finding points to a fundamental difference between LMICs and HICs in terms of how adolescents experience menstruation and where challenges lie in improving menstrual health literacy across contexts with varying economic and social resources.

### 3.2. Menstrual Health Issues

#### 3.2.1. Low and Middle-Income Countries (LMICs)

For girls in LMICs, pain or discomfort during periods was generally listed second after inadequate sanitary facilities as a cause for absenteeism from school [27,42,58,60]. Despite experiencing a variety of symptoms during menstruation such as pain, headaches and fatigue, studies conducted in LMICs found that very few girls sought help from medical professionals and tended to suffer silently from dysmenorrhea and discomfort [30,40,54,61], with study participants in Indonesia describing dysmenorrhea as a normal or natural occurrence with professional treatment only becoming necessary when the pain becomes unbearable [62]. They avoided a visit to the doctor on account of cultural taboos [62] or being uncomfortable with male doctors [63]. This consequently led to under-diagnosis and the delayed treatment of menstrual health disorders [40,58,64].

In a study in Uganda, girls had limited access to analgesics or pain-relief medicines due to the belief that they are not good for one’s health [47]. These students generally resorted to self-medication or home remedies [42,57,58,59,65] such as hot-water bottles or heating pads [64], spices such as garlic [43], or yoga [66]. Lack of access to effective pain medication in school also impacted the girls’ school performance [28] and thus warranted the need to make available pain-relief medicines in schools to bring relief from severe period pain or dysmenorrhea [29,36,60]. 

#### 3.2.2. High-Income Countries (HICs)

The 13 studies of HICs included in this review mainly discussed menstrual health issues amongst adolescents. Despite the existence of compulsory menstrual health education programs in schools in the majority of HICs, adolescents continued to suffer from high rates of dysmenorrhea and menstrual health disorders such as endometriosis that adversely impact their mental [67,68], physical [15,18,61,63,68,69,70,71,72] and social health [61,72]. Mental health issues such as depressive moods and psychological stress were found to be associated with menstrual irregularity and dysmenorrhea [67,68]. Absences from school and extra-curricular activities were not limited to adolescents diagnosed with endometriosis, but were also reported for those who suffered from moderate to severe dysmenorrhea in Sweden [69], Singapore [63], Switzerland [68], Finland [71], Australia [15] and Kuwait [73].

Despite HIC studies reporting a high incidence of menstrual pain, there was little evidence of early diagnosis and treatment of menstrual health disorders since adolescents did not actively seek medical or parental help [61,63,67,69,71] due to an underlying belief that their pain is a ‘normal’ part of their menstrual cycle [63,69]. These girls ended up resorting to self-management of pain via home remedies such as hot-packs and Chinese traditional medicine [68,73], and pain-relief medicines such as analgesics and Non-Steroidal Anti-Inflammatory Drugs (NSAIDs) [15,61,63,67].

The menstrual health issues of adolescents in both LMICs and HICs were similar in that adolescents in both groups tended to normalize menstrual pain and to seek alternative treatments rather than seek medical advice, unless the pain was extreme. Given the importance of early diagnosis of chronic conditions such as endometriosis, this widespread finding is concerning, although strategies to address the issue may need to be tailored for the different contexts. In LMICs, for example, the taboo of seeking medical advice from a male medical practitioner was a primary deterrent for many young women not found in HICs. 

### 3.3. Attitudes towards Menstruation

#### 3.3.1. Low and Middle-Income Countries (LMICs)

Menstrual experiences for adolescent girls in LMICs are shrouded in a culture of secrecy in both schools and families, coupled with embarrassment, shame and restrictions based on taboos, and consequently it is difficult to determine the actual lived experiences for these young women [28,31,47].

Effective menstrual practices by adolescents in schools are hindered by powerfully embedded cultural beliefs. Some cultures believe that menstrual blood is used by witches or Satanists to lure evil spirits into girls’ bodies [74] or negatively impact fertility [26], and this can impact the use and disposal of period products due to fear that their blood may be obtained from these. 

Given its association with reproduction, menstruation has been sexualized, which has consequentially led to it being made a taboo topic with families avoiding discussions to maintain ‘purity’ or ‘innocence’ of their children [74]. When a girl was menstruating, some communities imposed restrictions on her participation in household chores such as cooking, cleaning [75] as well as worshipping Gods in the temples [50], since they were considered ‘dirty’ or ‘polluting’ during this time [74]. The practice of Chhaupadi in Nepal, wherein menstruating women and girls are considered impure and deemed untouchable and are forced to live in exile with bare necessities, thus barred from participation in household activities, is one of the primary examples of deep-rooted cultural taboos having hazardous consequences on adolescents’ health [76]. No studies were found that offered interventions for these attitudes found within the home.

Girod et al. (2017) discuss how the onset of menstruation introduces new gendered inequities in school leading to cases of harassment and assault from boys, who tease girls and tell them to ‘go make a home’ or ‘go get married’, comments underpinned by the linkage being made between menstruation and the sexual aspect of reproduction [51,75]. The adolescent females are reportedly afraid of being ‘discovered’ since they are made to believe that their male counterparts are not supposed to know about menstruation and thus it is their duty to keep it a secret [77] and safely avoid the risk of harassment [78]. In Maasai culture, girls are prohibited from sharing a toilet with boys and prefer to use the nearby bushes around school [38]. In Cambodia [32] girls experience significant discomfort in using toilets located close to the boys’ toilets without a separation, allowing boys to observe their activities such as carrying a pad. Menstrual health and hygiene have become stigmatized issues that are not discussed openly in schools or at home [40,56] leading to a lack of awareness and preparation for menstruation [33,40] and psychological support for girls [35,40] at the time when they most need it.

While inclusion of boys in menstrual education is recommended, in China [56], boys and girls taking a menstrual education class together has been found to impede the discussion due to misbehavior and lack of seriousness on the part of the males. At the same time, a study conducted in India [78], with adolescent boys in schools, revealed that the limited information available to them about menstruation, due to a culture of secrecy, further perpetuated the stigma surrounding menstruation and menstrual issues. The study further discussed how boys were curious to gain more information, to the extent that they wanted menstrual education to become a part of their curriculum. The researcher also noted that one of the reasons behind their curiosity was also the change in attitude and/or the shift in the fundamental friendships between boys and girls, owing to the belief that girls are ready for marriage after menarche and thus imposing restrictions on female friendships [78]. This is an unfortunate belief that is not limited to one country, with similar observations being made in Zambia [26] and Kenya [51,75,78] where girls are being told to avoid ‘playing’ with boys since it puts them at risk of becoming pregnant.

Unfortunately, this belief is further perpetuated by teachers who subject students to judgmental teaching [79] and place the responsibility of not being harassed or assaulted on the adolescent females themselves, by asking them to be more careful and stay away from the boys if they wish to be safe, since now their friendliness could be misconstrued as a sexual advance [26,51]. Another issue faced by adolescent females is their negative encounters with male teachers who have a lack of understanding or interest towards their situation [33]. This lack of interest has also been cited as a cause for schools being unable to dedicate ideal levels of resources towards menstrual health and hygiene management [28]. Thus, adolescents in these countries prefer female teachers with adequate knowledge and confidence [43,56] to deliver menstrual health lessons [32,37]. Thus, the presence of a ‘female confidant’ teacher who forms the bridge between the school administration and female adolescents is essential for the success of a menstrual hygiene and management intervention in schools [80]. This, together with the stigma and fear of being teased by boys, teachers or other school staff [31] and the shame associated with accidental leakage and staining, was also found to be a cause of hindrance to school performance [27,28,44,47].

#### 3.3.2. High-Income Countries (HICs)

In HICs some families were found to avoid discussion of ‘female problems’, making it difficult for adolescents to seek the right support and treatment for menstrual health issues when most needed [69] and perpetuating negative stereotypes [72]. In Taiwan, menstruation is kept secret by female adolescents to avoid male taunting [61]. Such secrecy contributes to young boys’ lack of education about menstruation. This further creates male-dominated social stereotypes about menstruation and sexualizing the experience by tagging girls as ‘moody’, ‘weak’ and ‘distant’ when having their periods [61]. The lack of open discussion and appropriate support also aggravates the risk of girls developing a negative attitude towards menstruation by considering it an inconvenience, which further alleviates the risk of these adolescents developing a psychological disorder in the future [67]. There were no studies that addressed interventions for this issue.

This review reveals that attitudes to menstruation remain problematic in both LMICs and HICs, with studies revealing the persistence of negative attitudes leading to perverse consequences for young women. In LMICs, the detrimental attitudes appear to be more culturally entrenched in both home and school settings, potentially leading to greater social exclusion for young women in comparison to their HIC counterparts.

### 3.4. Limitations

While categorization of differences in menstrual health and menstrual hygiene by economic status of the country is common [13,81,82], it is important to note that this may overlook the relative contribution of various cultural or religious beliefs around menstruation in the cultural milieu of those countries [6], and it is important to acknowledge this review does not attempt to disentangle the various contributing factors such as culture, religion or geography that may contribute to this.

## 4. Conclusions

This review demonstrates that menstrual health literacy in low, middle and high-income countries is still inadequate and fails to cater to the needs of adolescents. Menstrual health is a cross-sectoral issue that requires a coordinated effort by the government and stakeholders in the education and health sectors [3]. Despite several interventions proving the success of an adequately designed program in improving the educational outcomes of adolescents regarding menstrual knowledge, there are limited pragmatic and policy responses to menstrual health literacy in adolescents [82].

Low health literacy is associated with poor overall health, incorrect/inadequate medication usage and lower levels of preventative health care [83]. More specifically, this review has highlighted that issues related to menstrual health literacy exist across LMICs and HICs, transcending geographic, cultural and socioeconomic status. There were more barriers for developing menstrual health literacy in LMICs, but there were also more overall studies of LMICs compared with HICs.

In LMICs, in addition to a lack of general knowledge about menstruation, compared with HICs, there are greater ‘practical’ burdens on young women when managing their menstrual health, leading to inadequate menstrual hygiene. These problems are related to greater absenteeism from school, increased lack of concentration and distress for young women at a crucial time in their educational development.

A general lack of knowledge about menstrual health issues was found by studies in both LMICs and HICs, commonly causing poor school performance, depressive modes, psychological stress, absenteeism and decreased participation in extra-curricular activities. Menstrual health issues were included in the school curriculum in HICs; however, there was little evidence that this was an effective solution for achieving better menstrual health literacy. Across both LMICs and HICs, menstrual health issues such as dysmenorrhea were normalized, leading young women to avoid seeking help from medical professionals or parents, and not taking advantage of analgesics or NSAIDs to minimize symptoms. In both LMICs and HICs, menstrual health is still associated with negative connotations leading to a lack of open communication both at school and at home.

This review found many interventions designed to assist with improving menstrual hygiene in LMICs, but fewer related to improving menstrual health knowledge. It is generally recommended that for interventions to be effective, they need to unite multiple stakeholders to align efforts across educational institutions, resources, infrastructure and attitudes. There is a significant gap in the literature in relation to interventions addressing dysmenorrhea in LMICs, but there is evidence of a high prevalence of menstrual health disorders. Similarly, there is a need for effective menstrual health programs in HICs to challenge the notion that period pain is ‘normal’, to educate about appropriate pain management and to assist with early diagnosis of menstrual health disorders. While access to educational resources and support groups appears effective in changing girls’ and teachers’ perceptions of menstrual health, there is a lack of studies illustrating effective ways to influence broader community attitudes.

Appropriate policies need to be developed to ensure evidence-based and cost-effective practical interventions that lead to an overall improvement in adolescent menstrual health. In the case of LMICs, there is a need for school-based menstrual health programs that provide both girls and boys with accurate information about menstruation. There is also a need to promote health discussions by challenging cultural taboos and providing girls with the necessary physical, mental, economic and infrastructural support to manage their menstrual experiences [3]. Additionally, there should be more research done examining menstrual hygiene in low-income areas of HICs. In the case of both LMIC and HICs, a consistent program that targets the early diagnosis and treatment of menstrual health disorders in adolescents is required such as the Menstrual Health and Endometriosis ‘me’ program in New Zealand and now, South Australian schools [84]. The LAL approach has seemed to be the most successful, and future efforts should work on using knowledge from various programs as well as funding for infrastructure improvements and resources. Interventions should be multifaceted, addressing most factors of menstrual health literacy and improving multiple parts of menstrual health literacy at once.

## Figures and Tables

**Figure 1 ijerph-18-02260-f001:**
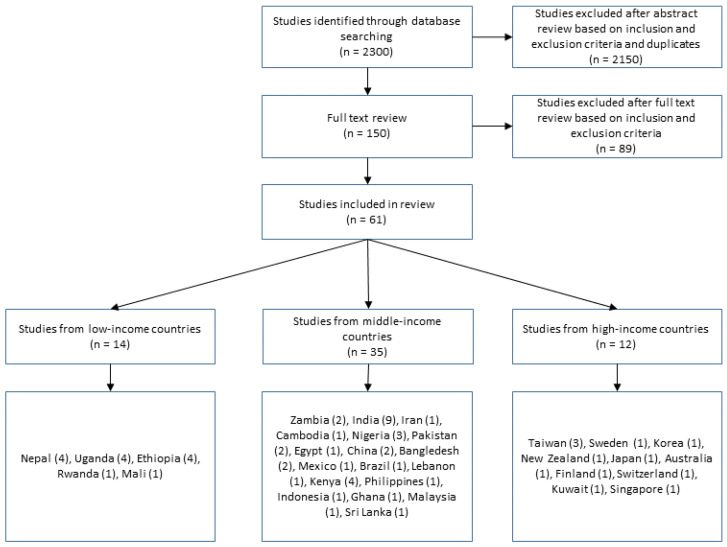
PRISMA flow chart of the search strategy.

**Table 1 ijerph-18-02260-t001:** Preliminary themes.

Inadequate Hygiene Practices and School Infrastructure	Lack of Information	Pain
No access to sanitary napkins Use of alternate material instead of disposable napkins WASH issues No management help *	Source of information: mother, sister, friend etc. No management information in school * Skipped the reproductive biology chapter	Fatigue Headaches Not going to doctor Home remedies Normalization Dysmenorrhea Diagnosis of disorders
**Cultural Taboos**	**Impact**	**Males**
Menstrual blood is dirty Menstruating females are impure Restricted participation in daily activities Keep it a secret * Becoming a woman *	Absenteeism * Anxiety * Embarrassment * Discomfort * Depression * Shame *	Sexualizing menstruation Prefer female teachers Harassment by boys Male teacher issue Keep it a secret * Becoming a woman *

* Indicates codes that are common across themes.

**Table 2 ijerph-18-02260-t002:** Final themes.

Menstrual Hygiene Management	Menstrual Health Issues	Attitudes Towards Menstruation
No access to sanitary napkins Use of alternate material instead of disposable napkins WASH issues No management information and help in school Source of information: mother, sister, friend etc. Skipped the reproductive biology chapter Absenteeism, anxiety, embarrassment and discomfort due to hygiene management issues	Fatigue Headaches Not going to doctor Home remedies Normalization Dysmenorrhea Diagnosis of disorders Absenteeism, discomfort and mental health issues due to menstrual health issues/disorders	Menstrual blood is dirty Menstruating females are impure Restricted participation in daily activities Keep it a secret Becoming a woman Sexualizing menstruation Prefer female teachers Harassment by boys Male teacher issue Absenteeism, anxiety, embarrassment, discomfort, depression and shame due to attitudes towards menstruation

## Data Availability

No new data were created or analyzed in this study. Data sharing is not applicable to this article.

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
