# Peer review of "Adolescent Menstrual Health Literacy in Low, Middle and High-Income Countries: A Narrative Review"

_ijerph, 2021, doi:10.3390/ijerph18052260_

Round 1

Reviewer 1 Report

The paper entitled "Adolescent Menstrual Health Literacy in Low, Middle and High-Income Countries: A Narrative Review” is very interesting and publishable in the journal. However, it requires major revision. My main concerns are: 1. The style of sentence structure is very poor. I suggest to the authors to consult with professional proofread service to correct the sentence structure throughout the paper. 2. In the first paragraph some information are not true. I have intensively work on this topic. In the western culture talking about the menstruation is not taboo topic. I suggest to authors to make correction and write exact information in the first paragraph of the introduction. 3. Line 52-54, how authors can state that the there is limited research literature on the proposed topic. It is not true. I can find 100s research papers on this issue/topic. 4. Line 77, it is not a formal way to write & in the heading. You must have to write and not &. 5. A lack of information or explanation related to Attitudes towards Menstruation particularly for developing and developed countries. Authors must have to add more literature to explain in an analytical way and it must be related to study topic. 6. Menstrual health issues have not been explained properly for low income countries. 7. Overall, I highly recommend to authors to please included a proper literature and explain it in an analytical way and conclude at the end in every explanation.

Reviewer 2 Report

This is a well-framed review on menstrual health literacy with appropriate methodology and selection strategy. The findings are well presented.

One limitation is a lack of discussion on the review itself, for example, any limitations come from grouping literatures by income, as the study found that cultural context plays a role?

Reviewer 3 Report

28-29 Eliminate the superscript meaning. This is a scientific journal. Ideologies must be exempted and set aside as far as science is concerned. Even more so when it is an ideology which attacks the biological foundations. Menstruation is an innate female biological process. Authors should know that gender identification is a social construct, an unscientific theory. 

Overall, there are a lot of typing errors, like empty spaces. With an easy over read of the authors should be fixed. 

In addition, authors should state the considerations for including the countries in either low, middle or high income. 

Some of the references do not correspond with the statement. eg 77 

Round 2

Reviewer 1 Report

The authors have addressed all the previous concerns. Now paper is publishable in its present form.